# Should I Help? Prosocial Behaviour during the COVID-19 Pandemic

**DOI:** 10.3390/ijerph192316084

**Published:** 2022-12-01

**Authors:** Walton Wider, Mei Xian Lim, Ling Shing Wong, Choon Kit Chan, Siti Sarah Maidin

**Affiliations:** 1Faculty of Business and Communications, INTI International University, Nilai 71800, Negeri Sembilan, Malaysia; 2Faculty of Health and Life Sciences, INTI International University, Nilai 71800, Negeri Sembilan, Malaysia; 3Faculty of Engineering and Quantity Surveying, INTI International University, Nilai 71800, Negeri Sembilan, Malaysia; 4Faculty of Data Science and Information Technology, INTI International University, Nilai 71800, Negeri Sembilan, Malaysia

**Keywords:** prosocial behaviour, COVID-19, moral elevation, moral judgement, moral identity

## Abstract

The Movement Control Order (MCO) enacted during the COVID-19 pandemic has profoundly altered the social life and behaviour of the Malaysian population. Because the society is facing huge social and economic challenges that need individuals to work together to solve, prosocial behaviour is regarded as one of the most important social determinants. Because it is related with individual and societal benefits, participating in prosocial activities may be a major protective factor during times of global crisis. Rather than focusing only on medical and psychiatric paradigms, perhaps all that is necessary to overcome the COVID-19 risks is for individuals to make personal sacrifices for the sake of others. In reality, a large number of initiatives proven to be beneficial in decreasing viral transmission include a trade-off between individual and collective interests. Given its crucial importance, the purpose of this concept paper is to provide some insight into prosocial behaviour during the COVID-19 period. Understanding prosocial behaviour during the COVID-19 pandemic is crucial because it may assist in the establishment of a post-COVID society and provide useful strategies for coping with future crises.

## 1. Introduction

In the first month of the year 2020, the SARS-CoV-2 (COVID-19) pandemic swept across the globe, and all nations instituted quarantine measures to prevent the virus from spreading further [1,2,3]. Starting on 8 March and continuing until 4 May, when it was partially lifted, Italy was the first European nation to implement a severe lockdown [4]. In addition, the government of Malaysia implemented the Movement Control Order (MCO) which is used when a pandemic becomes severe in the country [5,6,7]. This is done with the intention of reducing the likelihood that residents will become infected with a virus through direct interaction with one another. Although these safety measures were successful in halting the spread of the virus, they were responsible for a considerable disruption of the social and communal life in the regions that were affected by the outbreak [5,8,9]. As a result, the significance of prosocial behaviour becomes even more essential in times of crises, which is when a society is faced with significant social, economic, and political issues that require individuals to come together and work together to find solutions. During the COVID-19 pandemic, it is therefore of the utmost importance to gain an understanding of how to increase and encourage prosocial behaviour. Examples of prosocial behaviour include providing assistance, contributions, sharing, and comforting others [10]. The development of prosocial behaviour is an essential component of human socialization and has been linked to significant increases in mental health and well-being, both for those who are on the receiving end of beneficial acts and for those who perform them [1,11,12,13]. Prosocial behaviour is a multifaceted concept that can be broken down into many different subcategories. Some of these subcategories include compassion, caring, love, sympathy, empathy, altruism, and kindness [14].

Prosocial behaviours have been classified according to many different dimensions, such as whether they are spontaneous and informal, planned and formal, personal and impersonal, and the amount of effort required from the person providing the assistance [15,16]. Acts of generosity and assistance that are low in cost, such as sending an uplifting message, are typically straightforward and one-time deeds of kindness. As an illustration of a high-cost behaviour that requires perseverance, moral fortitude, and sustained commitment, emergency volunteer work is an option that may run counter to an individual’s own best interests [17,18]. From a psychological and social point of view, one explanation for prosocial behaviour is that people who help others feel a duty to do so [19]. At the community level, a sense of community responsibility (SOC-R), as coined by [20] has been conceptualized to represent a sense of duty toward others. Individuals are motivated to help one another and the community as a whole because they have a sense of obligation toward other members of the community as well as the community as a whole, which is what this phrase alludes to [20,21].

### 1.1. How Does Prosocial Behaviour Develop during Crises?

The term “catastrophe compassion” was coined to describe how people respond to widespread catastrophes by participating in acts of altruism toward others. This idea was devised to explain how people react to widespread disasters by engaging in acts of altruism [22]. These kind of selfless behaviors in the middle of widespread catastrophes are the consequence of individuals having similar social identities and an emotional connection with others who are going through the same kinds of challenges as they are [22,23]. An excellent approach to persuade people to commit to the in-group cause and drive them to behave in line with that commitment is to publicly demonstrate prosocial and unselfish behaviour [24]. Despite the fact that the COVID-19 pandemic posed a significant threat to the health of people all over the world and that its methods of containment were unprecedented, the pandemic was eventually brought under control. As a direct consequence of this, the crisis in question is unique, and it has compelled significant behavioural shifts, including changes in prosocial behaviour [24]. In addition, earlier studies have shown that exposure to similar calamity and anguish can lead to the development of more altruistic responses [25,26]. Cooperation was also demonstrated by musicians performing concerts from their balconies for the benefit of the community, and by individuals who clapped from their windows to show their appreciation to those working in frontline and healthcare roles [27]. In addition, in order to provide direct assistance to people who are in need, it was necessary for the collaborative efforts to be comprised of both organised volunteer efforts and informal community-based initiatives [28]. According to this point of view, having a traumatic experience creates a sense of shared fate and identity, which may lead to increased empathy and a greater desire to assist those who are in need of assistance.

### 1.2. Prosocial Behaviour during the COVID-19 Pandemic

During the COVID-19 pandemic, the majority of Canadians exhibited high levels of prosociality and compassion, as well as experienced those high levels themselves [1]. These findings are in line with those obtained in other nations including the United States of America, the United Kingdom, and Germany [29,30,31]. In point of fact, researchers have found that the public health guidelines developed in a country as a result of COVID-19 have created opportunities for increased empathy and prosociality toward other people [29,30]. For example, Andersson et al. found that people with high levels of prosociality are more likely to adhere to public health guidelines than people with low levels of prosociality [29]. These guidelines include things like keeping a social distance, wearing a mask properly, and always following hygiene practices in an effort to protect others from getting sick. This may be due to the fact that people who have higher levels of prosociality view interventions in public health as being acts of kindness performed towards other people.

During the COVID-19 pandemic, a grass-roots movement and various local initiatives emerged with the goal of providing assistance to populations that were at risk [32,33]. It is possible to define a grass-roots movement as the strategies that individuals in a specific district, region, or neighbourhood use as the foundation for political or economic movements [34]. This is analogous to the so-called “common-enemy effect”, which states that individuals are more likely to work together when confronted with a common foe, regardless of whether that foe is caused by nature, an individual, or a group [35,36]. Adversity indeed can bring out the best in people [32,37]. This finding demonstrates that prosocial behaviour will significantly increase during adversities such as the COVID-19 pandemic.

### 1.3. How Can We Foster More Prosocial Behaviour during the COVID-19 Pandemic?

#### 1.3.1. Moral Elevation

When we see the virtuous actions of another person, we feel a spectrum of pleasant moral emotions, including moral elevation as one of those feelings [38,39]. The idea of moral elevation refers to a collection of reactions that individuals have that inspire them to participate in prosocial behaviour and that are akin to other moral sentiments. This collection of responses is what motivates people to engage in prosocial behaviour [38,39]. Motives associated with moral elevation include the desire to become a better person and the urge to open one’s heart to others. The interrelated components, which include ideas, emotions, motives, and physical changes [40,41]. The aspiration to become a better person and the want to extend one’s heart to others are both desires that are associated with the process of moral elevation [42,43]. It was also demonstrated by Algoe and Haidt that moral elevation involves one’s own subjective feelings, such as a feeling of inspiration or elation, as well as one’s own bodily sensations [44]. In addition to this, holding views that are better from a moral standpoint presents a favourable image of mankind [45,46].

As is the case with other emotions related with morality, moral elevation has the ability to operate as a substantial source of moral inspiration and motivation, directing a person toward acts of goodness and deterring them from participating in immoral behaviour [47,48,49]. Numerous research has provided evidence that demonstrates the enormous effect that moral progress has on prosocial behaviour. Those who have reached a greater degree of moral development, for example, are more likely to assist others and to work toward life objectives that are more ethically oriented. Those who have not reached a higher level of moral development are less inclined to help others [44,50]. There is a link, as stated by Freeman et al., between exposing individuals to acts of moral goodness and increasing the quantity of charitable gifts they make [45]. As a direct consequence of this, moral development has come to be seen as a key driver of prosocial behaviour in humans [38,51]. People will participate in prosocial behaviour during the COVID-19 pandemic when their moral feelings assist to bridge the gap between moral principles and real behaviour [47,52]. Because of this, evoking a higher moral standard not only motivates prosocial behaviour when moral ideas are present, but it also adds to the accumulation of the effects of moral judgement and moral standard in order to inspire prosocial behaviour [53].

#### 1.3.2. Moral Judgement

The ability to generate moral judgments based on one’s own moral beliefs and the ability to conduct in line with those judgements is what is meant by the phrase “moral judgement competence” [54,55]. It is a measurement of how far a person has gone down the path of moral and cognitive growth over the course of their lifetime. According to Kohlberg, the maturation of an individual’s moral and cognitive capabilities takes place over the course of three separate levels, with each level including two phases [54]. These stages are representative of the level of moral reasoning that an individual reaches when they are able to differentiate between appropriate and inappropriate behaviours and are motivated to behave in a prosocial manner. When an individual reaches this level of moral reasoning, they are able to act in a manner that is beneficial to others. Because of the pivotal role that it plays in moral action, research on moral judgement has been carried out for a very long time. According to the theory of cognitive development outlined by [54], moral judgement may intrinsically motivate acts of prosocial behaviour. The growth of a person’s capacities for moral judgement will occur concurrently with the development of that person’s moral ideals. In addition, it is believed that individuals develop the capacity to use their moral beliefs as a foundation for decision making and to control the behaviour of both themselves and others via the cultivation of this skill. As a consequence of this, as individuals become older, their behaviours have a tendency to line more closely with the moral beliefs they’ve established. The results of research carried out by Ding et al. suggest that moral judgement is directly responsible for the motivation of prosocial behaviours such as moral elevation [53]. A person’s aptitude for moral judgement will play a part in deciding whether or not they are willing to engage in prosocial behaviour throughout the course of the COVID-19 epidemic.

#### 1.3.3. Moral Identity

Aquino and Reed explain that the relevance of being moral to the self may be shown by demonstrating that moral identity is built of a set of moral qualities [56]. The foundation for the trait-based definition of honesty that was proposed by Kingsford et al. is the argument that some moral characteristics, such as being kind or helpful, may be more fundamental to an individual’s self-concept than others, such as being honest or generous [57]. A person who considers themselves to be moral will, in order to maintain coherence between their self-definition and their actions, pursue more moral goals and exhibit more prosocial behaviour, such as offering assistance to others and showing concern for the emotions that they are experiencing. This is because a person who considers themselves to be moral will want to maintain coherence between their self-definition and their actions [46]. As a consequence of this, moral identity has also been believed to serve as a means of self-regulation that encourages moral conduct [8,46,57].

Numerous empirical investigations have shown that there is a strong and positive association between moral identity and prosocial behaviour [46,56]. In addition, the study indicated that those who had a strong sense of their own moral identity were more likely to participate in acts of charity and contribute more money than those who did not have a strong sense of their own moral identity [58,59]. As a consequence of this, the outcomes of this research suggest that moral identity might be used in order to explain moral behaviour [58]. Research carried out by Ding et al. indicates that having a moral identity might boost the impact of moral elevation, which in turn motivates prosocial behavior [53]. To summarise, when people have the same moral identity, it may encourage them to imitate the model, but when there is a large gap in moral identity between them, it may cause them to be hesitant to imitate prosocial behaviour. This can be seen in situations where there is a large gap in moral identity between people.

### 1.4. Why Is Prosocial Behaviour Crucial in Times of Crises?

To build a community that is resilient in the face of hardship, such as the COVID-19 epidemic, members of the community must engage in behaviours that benefit others. One definition of community resilience is a community’s continual capacity to tolerate, adapt to, and recover from misfortune. According to PeConga et al., prosocial behaviours increased community resilience by giving the act of surviving the crises a feeling of purpose and adaptive significance [60]. Communities, for example, have a duty to help individuals in times of greatest need by providing them with both monetary and emotional resources. This obligation exists in order for communities to carry out their role of assisting people. Tolerance, support, and generosity have the potential to serve as a buffer against the negative effects of a crisis situation [60,61]. There are several factors that contribute to a community’s resilience; nevertheless, two of them are likely to be substantial benefits of prosocial behaviour [62]. The first factor is having transformational community potential, which involves the notion of being able to analyse and grasp collective experiences in order to get access to and build community capacities to cope with such experiences [62,63]. The second factor includes perceptions of the community’s capacity to deal with catastrophes, including disaster preparation as well as community rehabilitation [23,62]. Furthermore, it has been demonstrated that engaging in prosocial behaviours may benefit the helper’s bodily and psychological well-being even when the beneficiaries are remote and not physically present. This is true even when the helper is assisting those who are not physically there, and it is true even when the beneficiary is not physically present [25,26,64,65]. These prosocial activities will also have a positive collective impact, such as increased opportunities for social contacts, solidarity, reciprocal support, and a feeling of being a competent and contributing part of the society. These cumulative impacts will benefit the community as a whole [23,66]. The aforementioned components are critical for the development of community resilience [67,68]. As a result, practising prosocial behaviours among community members is projected to increase views of the group’s capacity to manage during the COVID-19 pandemic. It has been shown that practising prosocial behaviours reduces feelings of isolation among community members because it promotes stronger perceptions of the community’s ability to cope [63,69].

At the level of the neighbourhood or the community, Stavrova and Schlosser mentioned that being prosocial can be the foundation of social equity and justice. Consequently, in order to emphasise the significance of prosocial behaviour during the COVID-19 epidemic, it is also essential to understand individuals who did not help, or even worse, those who caused harm to others [70,71]. This is reflected by a personality characteristic known as “justice sensitivity”, which may be described as the individual’s subjective perception of the relative importance that justice plays in their day-to-day existence [72,73]. Individuals with a high level of this trait witness injustices more often, have stronger emotional responses to them, and are more motivated to remedy or prevent them. For instance, certain vulnerable groups have been subjected to ethnic discrimination and environmental racism, both of which have contributed to the absence of prosocial behaviour that these groups have experienced throughout the epidemic. During the COVID-19 outbreak in China, the transport policies have disadvantaged certain population groups and exacerbated transport inequalities [74,75,76]. This is especially true when there are travel restrictions and public transport shutdowns. For instance, residents of non-gated communities were only allowed to engage in outdoor activities within their own neighbourhoods as a result of transit rules, which prevented them from participating in a variety of social and physical activities outside of their communities. This resulted in a significant increase in the levels of stress and depression among the populations that were already at a disadvantage [74]. In this particular instance, it has been demonstrated that a lack of prosocial behaviour may be particularly harmful to certain underprivileged populations, including having a negative impact on their health [77,78].

Ayalon also mentioned that age is a significant risk factor for COVID-19 mortality, and that older people are the ones who are suffering the most as a result of this pandemic [79,80]. For example, research has shown that younger generations frequently mock and scold older generations, particularly when the older generations are unable to regain their mobility despite the strictest travel prohibitions. This is especially true when younger generations see older generations as being unable to restore their mobility [79,81]. As a direct consequence of this, ageism and intergenerational conflict have become more prevalent around the world, to the point where senior citizens frequently face prejudice, discrimination, and stereotyping [82,83]. Aside from that, as was stated earlier, people who have a higher level of prosociality have a tendency to view public health measures as an act of kindness towards society. Research also highlights the fact that the older generation is seen as a threat to society because they are more self-centered and have a lower level of prosociality [84]. This is due to the fact that they typically do not adhere to the recommendations provided by the public health sector, such as keeping an appropriate social distance and refusing to wear face masks correctly [30,85]. However, other studies contend that the older generation was portrayed as a selfless and vulnerable group that was willing to jeopardize its own well-being in times of resource scarcity in order to benefit society as a whole [86]. This is a claim that is supported by the fact that the older generation is the largest demographic in the United States. For example, Dan Patrick, the lieutenant governor of Texas, has stated that he would prefer to die than to bring the nation’s economy to its knees [87]. These claims are in line with the findings of a previous study, which discovered that older people are typically expected to step down from their positions in favour of younger generations simply due to the chronological age gap that exists between the two groups [88,89].

### 1.5. Helping One Another Is Essential during the COVID-19 Pandemic

During the COVID-19 outbreak, the physical separation and quarantine rules severely disrupted people’s normal daily social contacts and promoted social isolation, which stands in stark contrast to the natural need that people have to interact with one another [90]. It has become more difficult to help other people as a result of the decline in the number of social contacts as well as the emergence of feelings of dread, insecurity, and tension [91]. This can lead to having concern only for oneself and contempt for those around you. In point of fact, it was found that the pandemic and behaviours associated with it caused a variety of responses that could be categorized as either self-centered or antisocial. These included issues such as overbuying and food stockpiling [92], which resulted in a lack of those goods and led to shortages. This school of thought contends that this loss of prosocial tendencies is caused by competition for those resources [32]. During times of crisis, people lose their prosocial tendencies due to struggles for scarce resources.

Research also highlights the importance of social norms in guiding people’s decisions about whether or not to act in a manner that is more self-interested or prosocial [93,94]. In times of widespread hardship, prosocial behaviour can be linked to benefits not only for the individual but also for society as a whole, which positions it as a factor that has the potential to serve as an effective protective buffer. For instance, if one complies with public health measures (such as wearing a mask), this will reduce the likelihood of the COVID-19 virus being passed on to others [94]. Acts that are done with the intention of helping other people are particularly important in light of the immense suffering that was caused by the COVID-19 pandemic. A study from the United States offers preliminary evidence that being prosocial during the COVID-19 pandemic increases positive affect, empathy, and social connectedness. This research shows that helping others has an immediate positive impact on the mood of the person doing the helping [63]. This finding is consistent with the phenomenon called the “common-enemy effect”, which holds that confronting a common enemy (such as one created by nature, a person, or a group) fosters cooperation because of the need to work together to defeat that enemy [35].

When we choose to tolerate the conditions during the COVID-19 pandemic that allow self-centeredness to flourish, we are making decisions that affect not only those who are affected by our self-centeredness at the time, but also the world we all live in and share, and self-centeredness, like the COVID-19 virus, spreads. To counter the impact of self-centeredness, one must do more than merely reject it. It entails accepting prosociality as a strategy of creating an environment that fosters health. In order to combat COVID-19, perhaps all that is necessary is for us to practice more prosocial behaviour. Rather than fighting COVID-19, we must embrace the interconnectedness that prosociality generates by saying: “we are all in this together”.

## 2. Conclusions

Engaging in prosocial behaviours can help to build community resilience, which can lead to the sustained ability of a community to withstand, adapt, and recover from adversity quickly. To increase the number of individuals engaging in prosocial behaviour during the COVID-19 pandemic, it is essential to promote moral elevation, moral discernment, and moral identity. Our assertion is congruent with Ding et al.’s conceptual model, which explores the underlying mechanism of morality on prosocial behavior [53]. Determining the mediating role of moral elevation on the relationship between moral judgement and prosocial behaviour, as well as the moderating influence of moral identity within the moral-self regulation system, is necessary to further study this conceptual paradigm during the COVID-19 pandemic. This is because each of these moral aspects has a distinct role in the system. In addition, this article discussed the various ways in which age and justice sensitivity influence the prosocial behaviours and moral motivations that people exhibit during COVID-19. As a result, we recommend that future researchers incorporate these two variables into the model developed by [53]. Figure 1 presents the proposed conceptual framework of the moderated mediation model.

## Figures and Tables

**Figure 1 ijerph-19-16084-f001:**
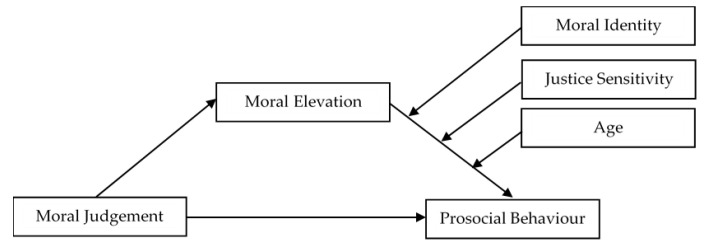
Conceptual model of the relationships between moral judgement, moral elevation, moral identity, justice sensitivity, age, and prosocial behaviour.

## Data Availability

Not applicable.

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
