# Peer review of "Should I Help? Prosocial Behaviour during the COVID-19 Pandemic"

_ijerph, 2022, doi:10.3390/ijerph192316084_

Round 1
Reviewer 1 Report
Dear Authors,
I am grateful for the opportunity to review your work. My comments and suggestions are as follows:
(1) I am unsure if I remember my grammar lessons well, but I would suggest that the title be rephrased as such: "Should I help? Prosocial behaviour during the COVID-19 pandemic".
(2) Perhaps the instance of MCO being mentioned in the abstract should include the fact that it originated from Malaysia. While this was seen in the body of the article, the initial impression I had was that this was from China.
(3) There was at least one instance that "behaviour" was spelt as "behavior". My suggestion is that we check the manuscript one last time for consistency.
(4) Given that this is an opinion piece, we might want to consider strengthening the transitions from and onto the different parts of the paper. One example: since the body of the article follows a question-and-answer format, the recommendations and judgments you make may be better reiterated in another section just before the conclusion.
(5) This is just me, but I have always been a firm believer that the first beneficiary of altruism is the person who demonstrates it. Perhaps this can be a jump-off point as well.
Again, thank you for the opportunity to read your work.
Author Response
Dear Examiner 1, we are grateful for your consideration of this manuscript, and we also very much appreciate your suggestions, which have been very helpful in improving the manuscript. All the comments we received on this manuscript have been taken into account in improving the quality.

Reviewer 2 Report
Thanks for inviting me to review this manuscript. This paper argues the importance of prosocial and reciprocal behaviour during a public crisis such as the Covid-19 pandemic. This is very interesting and the arguments are insightful. I think it is among the best manuscripts I have reviewed for this journal. Therefore I have only one major comment:
I think, to understand people who help each other, it is important to take a look at those who did not help, or even worse, those who hurt others. I believe this could be another strong point indicating why prosocial behaviour is crucial in times of crisis (the section before the conclusion). Because lacking prosocial behaviours may be especially harmful to certain disadvantaged populations (Montada & Schneider, 1989). And prosocial behaviour can be the bedrock of social equity and justice at neighbourhood or community level (Stavrova & Schlösser, 2015). For example, the intensified intergenerational tension and the conflict between younger and older people (Ayalon, 2020). Liu et al. (2021) found that older people cannot restore their mobility after the strictest travel restrictions were lifted because they were rebuked and humiliated by younger people. Other vulnerable populations may also suffer from the absence of prosocial behaviour. Liu et al. (2022) revealed the plight of discriminated-against migrant workers during the pandemic. They were discriminated against not only by local residents but other migrant workers and therefore could not sufficiently participate in activities within their neighbourhood.
Reference
Ayalon, L. (2020). There is nothing new under the sun: Ageism and intergenerational tension in the age of the COVID-19 outbreak. International Psychogeriatrics, 32(10), 1221-1224.
Liu, Q., Liu, Y., Zhang, C., An, Z., & Zhao, P. (2021). Elderly mobility during the COVID-19 pandemic: A qualitative exploration in Kunming, China. Journal of transport geography, 96, 103176.
Liu, Q., Liu, Z., Kang, T., Zhu, L., & Zhao, P. (2022). Transport inequities through the lens of environmental racism: rural-urban migrants under Covid-19. Transport policy, 122, 26-38.
Montada, L., & Schneider, A. (1989). Justice and emotional reactions to the disadvantaged. Social Justice Research, 3(4), 313-344.
Stavrova, O., & Schlösser, T. (2015). Solidarity and social justice: Effect of individual differences in justice sensitivity on solidarity behaviour. European Journal of Personality, 29(1), 2-16.
Author Response
Dear Examiner 2, we are grateful for your consideration of this manuscript, and we also very much appreciate your suggestions, which have been very helpful in improving the manuscript. All the comments we received on this manuscript have been taken into account in improving the quality.

Round 2
Reviewer 2 Report
Thanks for revising. I am satisfied with the current manuscript.
Author Response
Thank you Examiner for the acceptance.